

# Molecular characteristic of activin receptor IIB and its functions in growth and nutrient regulation in *Eriocheir sinensis*

Jingan Wang, Kaijun Zhang, Xin Hou, Wucheng Yue, He Yang, Xiaowen Chen, Jun Wang and Chenghui Wang

Key Laboratory of Freshwater Fisheries Germplasm Resources, Ministry of Agriculture and Rural Affairs, National Demonstration Center for Experimental Fisheries Science Education / Shanghai Engineering Research Center of Aquaculture, Shanghai Ocean University, Shanghai, China

## ABSTRACT

Activin receptor IIB *(ActRIIB)* is a serine/threonine-kinase receptor binding with transforming growth factor-β (TGF-β) superfamily ligands to participate in the regulation of muscle mass in vertebrates. However, its structure and function in crustaceans remain unknown. In this study, the *ActRIIB* gene in *Eriocheir sinensis* (*Es-ActRIIB*) was cloned and obtained with a 1,683 bp open reading frame, which contains the characteristic domains of TGF-β type II receptor superfamily, encoding 560 amino acids. The mRNA expression of *Es-ActRIIB* was the highest in hepatopancreas and the lowest in muscle at each molting stage. After injection of *Es-ActRIIB* double-stranded RNA during one molting cycle, the RNA interference (RNAi) group showed higher weight gain rate, higher specific growth rate, and lower hepatopancreas index compared with the control group. Meanwhile, the RNAi group displayed a significantly increased content of hydrolytic amino acid in both hepatopancreas and muscle. The RNAi group also displayed slightly higher contents of saturated fatty acid and monounsaturated fatty acid but significantly decreased levels of polyunsaturated fatty acid compared with the control group. After RNAi on *Es-ActRIIB*, the mRNA expressions of five *ActRIIB* signaling pathway genes showed that *ActRI* and forkhead box O (*FoxO*) were downregulated in hepatopancreas and muscle, but no significant expression differences were found in small mother against decapentaplegic (*SMAD*) 3, *SMAD4* and mammalian target of rapamycin. The mRNA expression s of three lipid metabolism-related genes (carnitine palmitoyltransferase 1β (*CPT1β*), fatty acid synthase, and fatty acid elongation) were significantly downregulated in both hepatopancreas and muscle with the exception of *CPT1*β in muscles. These results indicate that *ActRIIB* is a functionally conservative negative regulator in growth mass, and protein and lipid metabolism could be affected by inhibiting *ActRIIB* signaling in crustacean.

Corresponding author
Chenghui Wang,
wangch@shou.edu.cn

## INTRODUCTION

In vertebrates, the transforming growth factor-β (TGF-β) superfamily consists of a large number of structurally and functionally related cytokine subfamilies, including TGF-βs, bone morphogenetic proteins (BMPs), activins, and growth differentiation factors (GDFs), which regulate a series of biological processes, such as cell differentiation, muscle growth, and embryonic development (*Santibanez, Quintanilla & Bernabeu, 2011*; *Morikawa, Derynck & Miyazono, 2016*). The TGF-β family members exert their biological functions via two heteromeric complexes of transmembrane proteins, namely, type I and type II receptors (*TGFBR1* and *TGFBR2*, respectively), and activate the small mother against decapentaplegic (*SMAD*)-dependent or *SMAD*-independent signaling pathways (*Hata & Chen, 2016*; *Nickel, Ten Dijke & Mueller, 2018*). The activin receptor type IIB (*ActRIIB*), a serine/threonine kinase transmembrane receptor, can bind diverse members (ligands) of the TGF-β family, including *activins A*, *BMP-2*, *BMP-7*, *GDF-8* (myostatin) and *GDF-11* (*Sako et al., 2010*). *ActRIIB* regulates muscle growth, embryonic development, and reproduction in vertebrates (*Chen et al., 2015*; *Morvan et al., 2017*). In invertebrates, the conserved structure of *ActRIIB* was first identified in *Xenopus* in 1992; it plays an important role in the development of the TGF-β family (*Mathews, Vale & Kintner, 1992*; *Dyson & Gurdon, 1997*).

*ActRIIB* signaling, besides its effect on muscle growth, can also regulate (improve or inhibit) adipogenesis (*Bielohuby, 2012*; *Li et al., 2016*). Fat contents were reduced drastically in high-fat-diet mice by blockade of *ActRIIB* signaling (*Akpan et al., 2009*; *Koncarevic et al., 2012*), and the interference on *ActRIIB* improved lipid profiles and prevented hepatic fat accumulation compared with the mice fed with the same high-fat diet but treated with vehicle only (*Bielohuby, 2012*). Therefore, the inhibition of *ActRIIB* signaling has become a novel therapeutic approach in improving obesity and obesity-linked metabolic diseases (*Koncarevic et al., 2012*). This strategy induces changes in the expression patterns of fat metabolism-related genes in adipose tissues (*Koncarevic et al., 2012*; *Xin et al., 2019*). However, the molecular mechanism underlying *ActRIIB*-mediated metabolic process remains poorly understood in animals.

In crustaceans, the TGF-β superfamily members display multifunctional characteristics in regulating growth, metabolism, immune response, and appendage regeneration (*Klinbunga et al., 2018*; *Zhou et al., 2018*; *Shinji et al., 2019*; *Zhou et al., 2019*). In particular, myostatin, which is a negative growth regulator in vertebrates, is involved in complex and potentially multimodal actions in crustaceans (*Lee et al., 2015*; *Mykles & Medler, 2015*; *Abuhagr et al., 2016*; *Zhuo et al., 2017*; *Yue et al., 2020*). This condition demonstrates that the TGF-β superfamily members perform different biological actions in crustaceans and vertebrates. As an important receptor of myostatin, *ActRIIB* gene has become a powerfully therapeutic target to improve individual weight in vertebrates. However, its structure and functions in crustaceans remain unknown.

The Chinese mitten crab (*Eriocheir sinensis*) is an important economic crab species with huge aquaculture industry in China; however, it is considered an invasive species in Europe and North America (*Huang et al., 2015*; *Wang et al., 2018*). Recently, the mRNA structure

and expression of *TGFBR1* were reported in *E. sinensis*, and the functions of TGF-β-like signaling mediated by *TGFBR1* in molting-related muscle growth were elucidated in crustaceans (*Tian et al., 2019b*). As an important receptor of TGF-β superfamily members, the structure, expression, and biological actions of *ActRIIB* in regulating growth and muscle mass have not been identified in *E. sinensis*. Furthermore, in our recent studies, myostatin exhibited complex and multimodal actions in *E. sinensis* (*Yue et al., 2020*). As a member of the myostatin/*ActRIIB* signal pathway, the complex and multi-model actions of *ActRIIB* in *E. sinensis* must be characterized. In the present study, we first identified the structural characteristics and expression profiles of *ActRIIB* in *E. sinensis* (*Es-ActRIIB*) in various tissues during different molting stages. Then, we conducted RNA interference (RNAi) on *Es-ActRIIB* to investigate molting/growth characteristics, tissue compositions, and expression levels of metabolism-related genes in *E. sinensis*. This study aimed to characterize the *Es-ActRIIB* gene and determine the multifunctions of *ActRIIB* in crustacean.

## MATERIAL AND METHODS

### Animal and tissue collection

Healthy samples of juvenile *E. sinensis* with a mean weight of 8.38 ± 0.52 g, whole appendages, and good vitality were collected from the Aquatic Animal Germplasm Station of Shanghai Ocean University. The crabs were raised in a recirculation aquaculture system with the temperature maintained at 27 °C ± 0.5 °C and fed twice daily with commercial diets (*Yue et al., 2020*). To detect the expression profiles of *Es-ActRIIB*, we collected 11 different tissues of each crab, including the eyestalk, hepatopancreas, heart, gills (mixed front and back gills), stomach, intestine, walking leg muscle, claw muscle, pectoral muscle, thoracic ganglia, and epidermis, in the postmolt, intermolt, premolt and ecdysis stages, which were determined in accordance with the seta morphological characteristics of mandible described by *Chan, Rankin & Keeley (1988)*. All the collected tissue samples were flash-frozen in liquid nitrogen and subsequently stored at −80 °C for further analysis. Sampling procedures complied with the guidelines of the Animal Care and Use Committee of Shanghai Ocean University (SHOU-DW-2017021) on the care and use of animals for scientific purposes.

### RNA extraction and cDNA synthesis

The total RNA was extracted using the RNA iso-Plus (Takara, Japan) following the manufacturer's instructions. The concentration of extracted RNAs was examined by a spectrophotometer (Eppendorf BioSpectrometer® basic, Hamburg, Germany), and the RNA integrity was detected by 1% agarose gel electrophoresis. The RNAs with an OD 260/280 value ranging from 1.8 to 2.0 were used for cDNA synthesis. The templates used for full-length cDNA sequence cloning were synthesized by the SMARTer RACE 5′/3′ cDNA Kit Components (Clontech, USA). The templates used for quantitative real-time polymerase chain reaction (qRT-PCR) were synthesized by using the PrimeScript™ RT reagent Kit with gDNA Eraser (TaKaRa, Japan) in accordance with the manufacturer's instructions.

## Cloning of *Es-ActRIIB*

The primers for cloning of *Es-ActRIIB* were designed using Primer Premier 5.0 software based on transcriptome annotation results of *E. sinensis* in our previous study (*Huang et al., 2015*) (Table S1). The total volume of the PCR reaction was 20 µL, including 10 µL 2×Hieff® Master Mix, 1 µL cDNA template, 0.5 µL 10 pmol/µL forward primer and 0.5 µL 10 pmol/µL reverse primer, and 8 µL double-distilled water (ddH$_2$O). The PCR programs were run as follows: 94 °C for 5 min; 30 cycles of 94 °C for 30 s, 60 °C for 30 s; 72 °C for 1 min; then 72 °C for 7 min. The PCR products were examined using 1% agarose gel. Then, the purified PCR products were ligated with pMD19-T Vector (TaKaRa, Japan) and transformed into *Escherichia coli* DH5α competent cell (TaKaRa, Japan). The positive colonies were sent to Sangon Biotech Company (Shanghai, China) for sequencing.

## Bioinformatics analysis of *Es-ActRIIB*

The protein sequence of Es-ActRIIB was deduced by ExPASY-translate tool (http://www.expasy.org/). The sequence alignment analysis was performed using the National Center for Biotechnology Information–Basic Local Alignment Search Tool (http://www.ncbi.nlm.nih.gov/blast). The protein domains were predicted by Simple Modular Architecture Research Tool (http://smart.embl-heidelberg.de/) and modeled by Iterative Threading ASSEmbly Refinement server (https://zhanglab.ccmb.med.umich.edu/I-TASSER/). A phylogenetic tree was constructed by MEGA 7.0 software using neighbor-joining methods with a bootstrap value of 1,000 (Table S2).

## Tissue expression detection of *Es-ActRIIB*

The mRNA expression levels of *Es-ActRIIB* in 11 tissues in the four molting stages were detected using 10 µL Hieff UNICON® qPCR SYBR Green Master Mix, including 1 µL cDNA, 1 µL primer Mix, and 8 µL ddH$_2$O. Four biological replicates and three experimental replicates were employed in the detected each tissue from each molting stage. The reaction program was run as follows: 95 °C for 30 s, followed by 40 cycles of 95 °C for 5 s and 60 °C for 30 s. The temperature was increased by 0.5 °C per 5 s from 60 °C to 95 °C for the melting curve with 30 s elapse time per cycle. *β-Actin*, *S27*, and *UBE* were selected as reference genes (*Huang et al., 2017*), and the relative expression levels of target genes were accurate normalized by geometric averaging of this three reference genes with the method of ΔΔCt (*Vandesompele et al., 2002*; *Hellemans et al., 2007*). The primers for qRT-PCR were listed in Table S1.

## RNAi

The target segment containing numerous functional short-interfering RNA sites was predicted using siDirect version 2.0 (http://sidirect2.rnai.jp/) to obtain an effective double-stranded RNA (dsRNA) of *Es-ActRIIB*. The designed primers included a T7 RNA polymerase-binding site at the 5′-end (Table S1). The dsRNA was synthesized using T7 RiboMAX™ Large Scale RNA Production Systems (Promega, P1300) in accordance with the manufacturer's instructions. To test the efficiency of the designed dsRNA, we injected six individuals with 3 µg/g dsRNA (1 µg/mL) as the experimental group and injected another six individuals with phosphate-buffered saline (PBS) (3 µL/g) as the control

group. qRT-PCR was conducted to test the expression of *Es-ActRIIB* for investigating the interference efficiency. Sixty individual crabs ($4.62 \pm 0.78$ g) were collected immediately after molting and divided randomly and equally into two groups, namely, the RNAi (injected with dsRNA) and control (injected with PBS) groups, to explore the function of *Es-ActRIIB* in growth during the molting process. All crabs were individually raised in the same recirculation aquaculture system with the temperature maintained at $27\,°C \pm 0.5\,°C$ and fed twice daily with commercial diets. The first injection was conducted on the 5th day after crab molting and continued every 5 days until the end of the second molting. At the end of the experiment, each survivor was weighed, sacrificed, and then dissected for analysis of gene expression and tissue composition.

## Basic growth characteristic measurement

According to our previous study (*Yue et al., 2020*), the body weight (*BW*) and hepatopancreas weight (*HW*) of each crab were measured on the third day after molting, and molting interval time (*MI*, the days between the first and second molting) was recorded for each individual. The weight gain rate (*WGR*), specific growth rate (*SGR*), and hepatopancreas index (*HI*) were respectively calculated as follows:

$$WGR = (BW_2 - BW_1) \,/\, BW_1 \times 100\%$$
$$SGR = (\mathrm{Ln}BW_2 - \mathrm{Ln}BW_1) \,/\, MI$$
$$HI = (HW \,/\, BW_2) \times 100\%$$

where $BW_1$ and $BW_2$ are the *BW* after the first and second molting, respectively.

## Tissue composition analysis

At the end of the RNAi experiment, the hepatopancreatic and whole muscle tissues of each crab were sampled for fatty acid and hydrolytic amino acid composition analysis by using gas chromatography and mass spectrometry and amino acid automatic analysis apparatus, respectively (*Wei et al., 2018*).

## Expression detection of target genes

Similarly, at the end of the RNAi experiment, six crabs from the RNAi and control groups were sampled, and their hepatopancreas and leg muscle tissues were collected for qRT-PCR detection of the target genes, including five *ActRIIB* signaling pathway genes (*ActRI*, *SMAD3*, *SMAD4*, forkhead box O (*FoxO*), and mammalian target of rapamycin (*mTOR*)) (*Han et al., 2013*; *Guru et al., 2019*), and three lipid metabolism-related genes (carnitine palmitoyltransferases 1β (*CPT1 β*), fatty acid synthase (*FAS*), and fatty acid elongation (*FAE*)) (*Liu et al., 2016*; *DeBose-Boyd, 2018*; *Liu et al., 2018*). The qRT-PCR procedures were similar to those for *Es-ActRIIB* expression detection. Table S1 lists all the primers for the qRT-PCR reactions.

## Statistical analysis

All data about gene expression, growth characteristic, and tissue composition were statistically analyzed by one-way analysis of variance followed by Duncan's multiple range test with SPSS 20. $P < 0.05$ was considered statistically significant.

## RESULTS

### cDNA sequence of *Es-ActRIIB*

The full-length cDNA sequence of *Es-ActRIIB* was 4916 bp (GenBank accession number: MN832896), including a 618 bp 5′ terminal untranslated region (5′-UTR), a 1,683 bp of open reading frame (ORF), and a 2,615 bp 3′-UTR. The ORF encodes 560 amino acids (aa) with a predicted molecular weight of 62.86 kDa and a theoretical isoelectric point of 6.00 (Fig. S1). The predicted domains of Es-ActRIIB contain an activin receptor domain (46–140 aa) and a serine/threonine protein kinase domain (251–539 aa), which are characteristic domains of TGFBR2 superfamily. A signal peptide and a transmembrane region were located at the 1–28 aa and 198–220 aa, respectively (Fig. 1A). The tertiary structure of Es-ActRIIB consists of 13 $\alpha$-helices and 7 β-sheets (Figs. 1B and 1C).

### Sequence alignment and phylogenetic analysis

BLAST analysis showed that Es-ActRIIB protein sequence shared high identities with *Portunus trituberculatus* (92%; GenBank: MPC26231.1), *Penaeus vannamei* (69%; GenBank: ROT74806.1), *Daphnia magna* (54%; GenBank: JAL80963.1) and *Nasonia vitripennis* (GenBank: XP_001603863.1) and other insects (50%–60%). The phylogenetic tree of ActRIIB could be divided into two large branches. ActRIIBs from arthropods were clustered together in one large branch, in which Es-ActRIIB was clustered closely with ActRIIBs of crustaceans. ActRIIBs of fishes, amphibians, mammals, reptiles and bird species were clustered together in another branch (Fig. 2).

### Tissue expression profiles of *Es-ActRIIB*

The mRNA expression of *Es-ActRIIB* could be detected in all 11 tissues during the four molting stages (Fig. 3). Although significant differences were observed among tissues at the same molting stage, the mRNA expression of *Es-ActRIIB* in hepatopancreas was the highest ($P < 0.05$) and the lowest in muscles (walking leg muscle, claw muscle, pectoral muscle) at each molting stage. Comparison of the four molting stages revealed the highest expression ($P < 0.05$) of *Es-ActRIIB* gene was detected in hepatopancreas and three muscle tissues at the premolt stage.

### Growth characteristics after RNAi on *Es-ActRIIB*

After injection of synthesized dsRNA, the mRNA expression of *Es-ActRIIB* significantly decreased by 60.84% ($P < 0.01$) in the RNAi group compared with the control group at 48 h (Fig. 4), demonstrating the high efficiency of the designed dsRNA on *Es-ActRIIB*. After continuous injection during one molting cycle, the *WGR* increased by 53.55%, and the *SGR* increased by 64.71% in the RNAi group ($P < 0.01$) compared with the control group. In addition, the *HI* in the RNAi group (5.26 ± 0.58) was significantly lower than that of the control group (7.09 ± 0.55) ($P < 0.01$). No significant difference was found in the *MI* between the RNAi and control groups ($P > 0.05$) (Table 1).

### Tissue composition changes after RNAi on *Es-ActRIIB*

After *Es-ActRIIB* dsRNA injection was continually conducted during one molting cycle, the contents of total essential amino acid (TEAA), total non-essential amino acid (TNEAA)

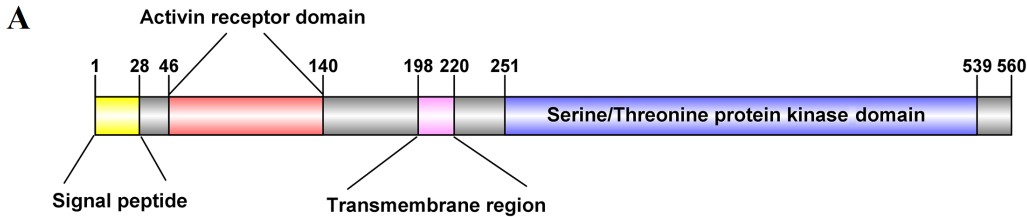

**A**

Activin receptor domain

1  28  46  140  198  220  251  539  560

Serine/Threonine protein kinase domain

Signal peptide

Transmembrane region

**B**

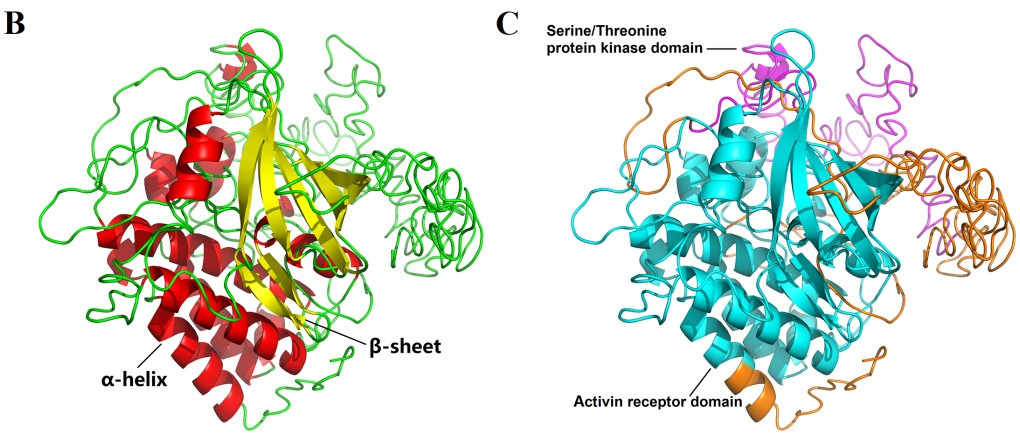

α-helix

β-sheet

**C**

Serine/Threonine
protein kinase domain

Activin receptor domain

**Figure 1** **Schematic diagram for structure prediction of Es-ActRIIB.** (A) Predicted domain structure of Es-ActRIIB. (B–C) Predicted protein structure of Es-ActRIIB. The $\alpha$-helix (red) and the $\beta$-sheet (yellow) were highlighted. The activin receptor domain (cyan) and the Serine/Threonine protein kinases domain (magenta) were highlighted.

and total amino acid (TAA) were significantly higher in the hepatopancreas and muscle tissues of the RNAi group than those of the control group (Table 2), demonstrating protein accumulation after the inhibition of *Es-ActRIIB*. In hepatopancreas, the RNAi group exhibited significantly higher contents of saturated fatty acid and monounsaturated fatty acid but significantly lower content of polyunsaturated fatty acid compared with the control group. For the muscle, the two groups displayed the same trends for the hepatopancreas and showed no statistically significant difference ($P > 0.05$) (Table 3).

## Expression changes of target genes after RNAi on *Es-ActRIIB*

The mRNA expression changes of related target genes were observed after *Es-ActRIIB* dsRNA injection for one molting cycle (Fig. 5). For *ActRIIB* pathway-related genes, the mRNA expressions of *ActRI* and *FoxO* were significantly downregulated in hepatopancreas and muscle ($P < 0.05$), but no significant expression differences were found in the other three genes (*SMAD3*, *SMAD4*, and *mTOR*) ($P > 0.05$). For lipid metabolism-related genes, the mRNA expression of *CPT1β* was significantly downregulated in hepatopancreas but upregulated in muscle ($P < 0.05$). Meanwhile the mRNA expressions of *FAS* and *FAE* were significantly downregulated in both hepatopancreas and muscle tissues in the RNAi group compared with the control group ($P < 0.05$).
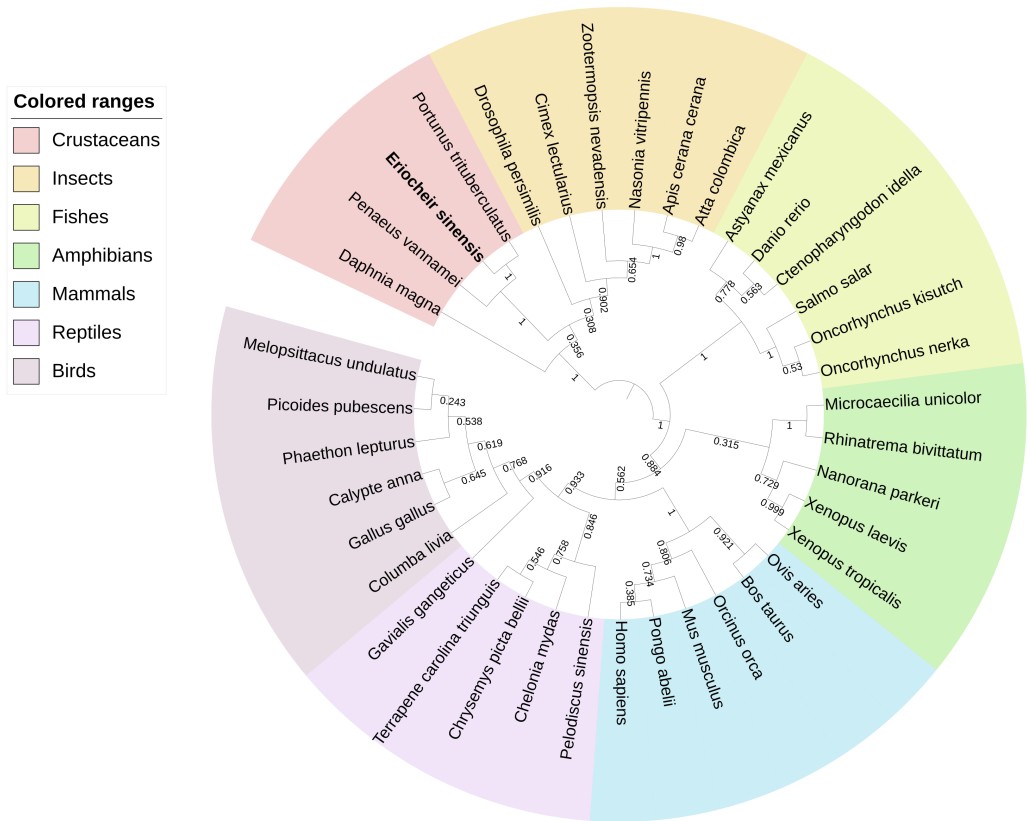

**Figure 2   Phylogenetic tree based on ActRIIB protein sequences.**

## DISCUSSION

In the present study, characterization of *Es-ActRIIB* indicated that the encoded protein sequence of this gene has the characteristic structural domains of TGFBR2 members (*Thompson, Woodruff & Jardetzky, 2003*). In vertebrates, *ActRIIB* binds to activin and myostatin ligands through its extracellular activin receptor domain, and its intracellular serine/threonine protein kinase domain phosphorylates down-steam *SMAD* signaling factors to exert signal transduction (*Shi & Massagué, 2003*). These results demonstrate that the domain is highly conserved in *E. sinensis*, similar to that in vertebrates, implying that *Es-ActRIIB* might play the same role in TGF-β signal transduction as in vertebrates.

   *ActRIIB* is widely distributed in various tissues and developmental stages in mouse, zebrafish, and other vertebrates (*Garg et al., 1999*; *Rebbapragada et al., 2003*), consistent with our findings showing that *Es-ActRIIB* was widely expressed in all tested tissues at different molting stages; the findings also indicate the similar expression profiles in crustaceans and vertebrates. Molting is the special growth biological process of *E. sinensis*, and the transcription levels of numerous related genes *in vivo* change with molting cycle (*Huang et al., 2015*). The expression profiles of *Es-ActRIIB* in different tissues showed significant changes in the four molting stages, revealing that *ActRIIB* is involved in molting-related growth regulation in *E. sinensis*. Interestingly, *Es-ActRIIB* showed the

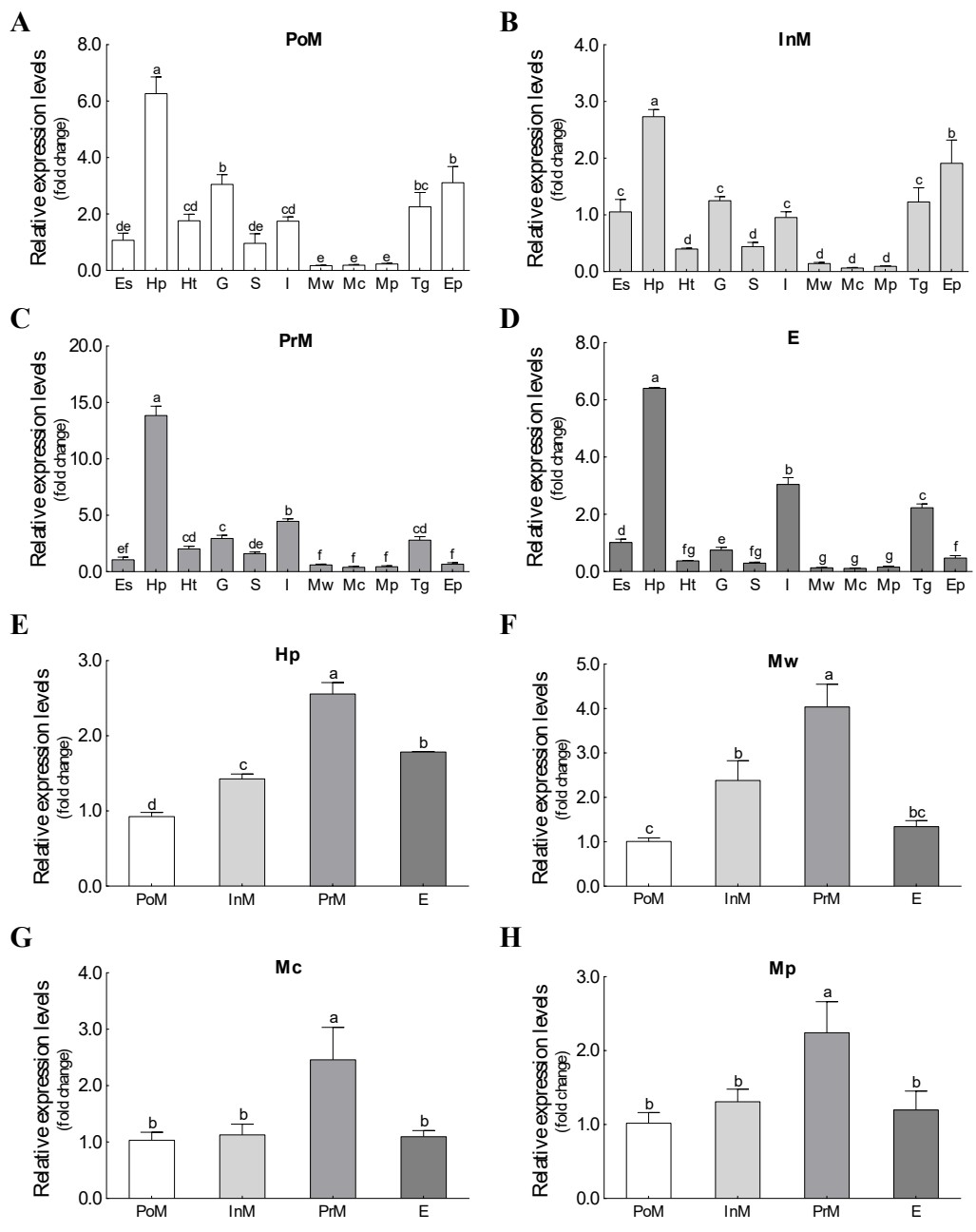

**Figure 3** **The expression profiles and comparison of *Es-ActRIIB* in different tissues and molting stages.** (A–D) The expression profiles of the 11 tissues at four molting stages, respectively. (E–H) Relative expression comparison among the four molting stages for relative tissues. PoM, postmolt stage; InM, intermolt stage; PrM, premolt stage; E, ecdysis stage; Es, eyestalk; Hp, hepatopancreas; Ht, heart; G, gill; S, stomach; I, intestine; Mw, walking leg muscle; Mc, claw muscle; Mp, pectoral muscle; Tg, thoracic ganglia; Ep, epidermis. Histogram plotted with mean and standard error. Different letters showed significant differences ($P < 0.05$).

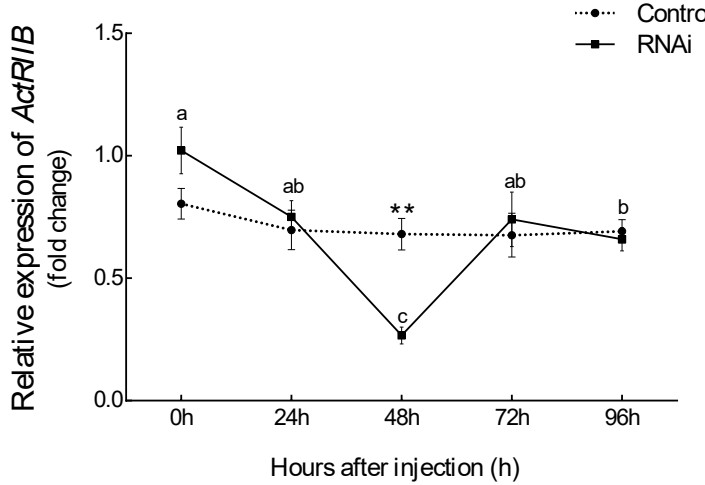

**Figure 4 The interference efficiency of designed *Es-ActRIIB* dsRNA.** The different letters (a, b and c) indicated significant differences between hours after injection; "**" indicated extremely significant difference between two groups ($P < 0.01$).

**Table 1 Effects of RNAi on growth performance of juvenile mitten crab (means ± SD).**

| Indicators of growth | RNAi ($n = 14$) | Control ($n = 14$) |
|---|---|---|
| $BW_1$ (g) | $4.72 \pm 0.86$ | $4.54 \pm 0.85$ |
| $BW_2$ (g) | $6.76 \pm 1.16$ | $5.86 \pm 1.28$ |
| WGR (%) | $43.87 \pm 8.23$[**] | $28.57 \pm 8.60$ |
| MI (d) | $44.79 \pm 7.53$ | $50.14 \pm 7.44$ |
| SGR (%/d) | $0.84 \pm 0.26$[**] | $0.51 \pm 0.16$ |
| HI (%) | $5.26 \pm 0.58$ | $7.09 \pm 0.55$[**] |

**Notes.**
[*]Significant difference ($P < 0.05$).
[**]Extremely significant difference ($P < 0.01$).
$BW_1$, body weight after the first molt; $BW_2$, body weight after the second molt; WGR, weight gain rate; SGR, specific growth rate; MI, molting interval time; HI, hepatosomatic index.

highest expression in the hepatopancreas in the four molting stages; this finding might be relevant to growth metabolism because hepatopancreas is an important organ underlying carbohydrate and lipid metabolism, nutritional status, energy storage, and breakdown in crustaceans (*Wang et al., 2008*). Meanwhile, the mRNA expression of *Es-ActRIIB* in all three types of muscles peaked at the premolt stage during the molting process, implying that this gene is involved in muscle atrophy induced by molting.

To explore the function of *Es-ActRIIB* in growth regulation, we successfully knocked down the transcription level of *ActRIIB* in *E. sinensis*. The *Es-ActRIIB* mRNA knocked-down crabs in this study showed fast weight gain rate and specific growth rate. This finding confirms the result of RNAi on *ActRIIB* in dystrophic mdx mice (*Dumonceaux et al., 2010*). Moreover, the high contents of TEAA, TNEAA and TAA in muscle and hepatopancreas indicated that the fast growth of *Es-ActRIIB* knocked-down individuals was caused by the acceleration of protein synthesis. The myostatin/*ActRIIB* signal pathway regulating

**Table 2  Amino acid composition in hepatopancreas and muscle of *E. sinensis* (g/100 g, dry weight).**

| Amino acid | Hepatopancreas | | Muscle | |
|---|---|---|---|---|
| | RNAi | Control | RNAi | Control |
| Asp | 2.70 ± 0.02[**] | 2.49 ± 0.04 | 3.89 ± 0.13[*] | 3.52 ± 0.13 |
| Thr[A] | 1.23 ± 0.01[**] | 1.13 ± 0.02 | 1.78 ± 0.07[*] | 1.60 ± 0.04 |
| Ser | 1.04 ± 0.01[**] | 0.92 ± 0.01 | 1.69 ± 0.05[*] | 1.54 ± 0.05 |
| Glu | 2.92 ± 0.01[**] | 2.68 ± 0.05 | 5.49 ± 0.14[*] | 5.02 ± 0.18 |
| Gly | 1.27 ± 0.01[**] | 1.16 ± 0.01 | 2.16 ± 0.08 | 2.12 ± 0.05 |
| Ala | 1.25 ± 0.01[**] | 1.12 ± 0.02 | 2.39 ± 0.08[*] | 2.23 ± 0.06 |
| Val[A] | 1.29 ± 0.02[**] | 1.15 ± 0.02 | 2.01 ± 0.08 | 1.89 ± 0.05 |
| Met[A] | 0.35 ± 0.10 | 0.28 ± 0.20 | 0.66 ± 0.15[**] | 0.24 ± 0.04 |
| Ile[A] | 0.88 ± 0.02 | 0.83 ± 0.04 | 1.67 ± 0.02[**] | 1.54 ± 0.04 |
| Leu[A] | 1.75 ± 0.01[**] | 1.61 ± 0.03 | 2.77 ± 0.06[**] | 2.54 ± 0.06 |
| Tyr | 1.18 ± 0.01[**] | 1.10 ± 0.01 | 1.68 ± 0.10 | 1.50 ± 0.13 |
| Phe[A] | 1.24 ± 0.04[*] | 1.14 ± 0.03 | 1.49 ± 0.10 | 1.36 ± 0.15 |
| His[A] | 0.60 ± 0.02[*] | 0.56 ± 0.00 | 1.13 ± 0.03[*] | 1.03 ± 0.05 |
| Lys[A] | 1.58 ± 0.00[**] | 1.45 ± 0.03 | 2.59 ± 0.09[*] | 2.34 ± 0.09 |
| Arg | 1.58 ± 0.02[*] | 1.54 ± 0.02 | 3.45 ± 0.14[*] | 3.17 ± 0.08 |
| Pro | 1.11 ± 0.02 | 1.05 ± 0.06 | 2.19 ± 0.10[*] | 2.02 ± 0.01 |
| TEAA | 8.93 ± 0.08[**] | 8.15 ± 0.22 | 14.10 ± 0.35[*] | 12.53 ± 0.45 |
| TNEAA | 13.06 ± 0.05[**] | 12.04 ± 0.11 | 22.95 ± 0.79[*] | 21.14 ± 0.62 |
| TEAA/TNEAA | 0.68 ± 0.01 | 0.68 ± 0.02 | 0.62 ± 0.01[**] | 0.59 ± 0.01 |
| TAA | 21.99 ± 0.11[**] | 20.19 ± 0.33 | 37.06 ± 1.13[*] | 33.67 ± 1.10 |

**Notes.**

[*]Significant difference ($P < 0.05$).

[**]Extremely significant difference ($P < 0.01$).

A, essential amino acid; TEAA, total essential amino acid; TNEAA, total non-essential amino acid; TAA, total amino acid.

muscle protein balance has been identified in vertebrates (*Han et al., 2013*). *ActRI* is the interacting protein of *ActRIIB* binding to myostatin (*Hata & Chen, 2016*), and its lower expression was observed with interference of *Es-ActRIIB* in this study; the expressions of downstream transcription factors *SMAD3* and *SMAD4* were not influenced, the reason probably was that *SMAD*s, as core and versatile cytokines, are active in the TGF-β pathway, including the signal transduction of TGF-β/BMP/activin, not only in myostatin/*ActRIIB* pathway (*Massague, Seoane & Wotton, 2005*; *Tian, Jiao & Cheng, 2018b*). Moreover, in the process of muscle protein balance, *ActRIIB* signal pathway stimulates *FoxO*-dependent transcription to enhance muscle protein catabolism and suppresses *Akt/mTOR* signaling to inhibit muscle protein synthesis (*Han et al., 2013*; *Hulmi et al., 2013*; *Tian et al., 2018*; *Tian, Lin & Jiao, 2019*). Although the expression of *mTOR* was not influenced after dsRNA injection of *Es-ActRIIB*, the low expression of *FoxO* was observed, resulting in the slowed down protein catabolism that is needed to accelerate muscle protein synthesis. This result was also consistent with the high TEAA, TNEAA and TAA contents, rapid weight gain rate, and specific growth rate. Therefore, the results imply that the interference of *Es-ActRIIB* accelerated individual growth by regulating protein catabolism and synthesis pathways.

**Table 3  Fatty acid composition in hepatopancreas and muscle of *E. sinensis* (% of total fatty acids).**

| Fatty acid | Hepatopancreas | | Muscle | |
|---|---|---|---|---|
| | RNAi | Control | RNAi | Control |
| C14:0 | $1.33 \pm 0.02$ | $1.40 \pm 0.01$[**] | $1.30 \pm 0.05$ | $1.37 \pm 0.03$ |
| C15:0 | $0.61 \pm 0.01$ | $0.64 \pm 0.00$[**] | $0.60 \pm 0.04$ | $0.59 \pm 0.08$ |
| C16:0 | $22.34 \pm 0.19$[*] | $21.56 \pm 0.25$ | $22.68 \pm 0.49$ | $21.86 \pm 0.43$ |
| C17:0 | $0.37 \pm 0.01$ | $0.38 \pm 0.01$ | $0.44 \pm 0.06$ | $0.47 \pm 0.04$ |
| C18:0 | $2.83 \pm 0.03$ | $2.79 \pm 0.04$ | $3.32 \pm 0.20$ | $3.71 \pm 0.32$ |
| C20:0 | $0.24 \pm 0.00$[*] | $0.23 \pm 0.01$ | —— | —— |
| C21:0 | $0.24 \pm 0.01$ | $0.23 \pm 0.01$ | —— | —— |
| C22:0 | $0.25 \pm 0.01$ | $0.24 \pm 0.00$ | —— | —— |
| C23:0 | $0.36 \pm 0.03$ | $0.40 \pm 0.03$ | $0.94 \pm 0.23$ | $1.16 \pm 0.30$ |
| C24:0 | $0.15 \pm 0.01$ | $0.16 \pm 0.00$ | —— | —— |
| **SFA** | **$28.73 \pm 0.09$**[*] | **$28.02 \pm 0.26$** | **$29.28 \pm 0.21$** | **$29.16 \pm 0.12$** |
| C14:1 | $0.21 \pm 0.01$ | $0.22 \pm 0.01$ | —— | —— |
| C16:1 | $9.80 \pm 0.04$[**] | $9.54 \pm 0.05$ | $8.99 \pm 018$ | $8.63 \pm 0.61$ |
| C17:1 | $0.63 \pm 0.01$ | $0.68 \pm 0.00$[**] | $0.54 \pm 0.01$ | $0.66 \pm 0.03$[**] |
| C18:1n9t | $0.18 \pm 0.02$ | $0.23 \pm 0.04$ | —— | —— |
| C18:1n9c | $32.47 \pm 0.28$[**] | $30.81 \pm 0.33$ | $31.74 \pm 0.16$[*] | $30.39 \pm 0.66$ |
| C20:1n9 | $0.88 \pm 0.00$[**] | $0.84 \pm 0.01$ | $0.87 \pm 0.02$ | $0.94 \pm 0.04$ |
| C22:1n9 | $0.12 \pm 0.00$ | $0.16 \pm 0.00$[**] | —— | —— |
| C24:1n9 | $0.45 \pm 0.02$[**] | $0.35 \pm 0.03$ | —— | —— |
| **MUFA** | **$44.73 \pm 0.23$**[**] | **$42.82 \pm 0.37$** | **$42.15 \pm 0.33$** | **$40.62 \pm 1.21$** |
| C18:2n6c | $21.63 \pm 0.05$ | $23.58 \pm 0.37$[**] | $22.95 \pm 0.16$ | $23.29 \pm 0.48$ |
| C18:2n6t | $0.12 \pm 0.00$ | $0.12 \pm 0.00$ | —— | —— |
| C18:3n3 | $1.82 \pm 0.07$ | $2.22 \pm 0.11$[**] | $2.10 \pm 0.06$ | $2.39 \pm 0.06$[**] |
| C18:3n6 | $0.37 \pm 0.02$[**] | $0.30 \pm 0.00$ | —— | —— |
| C20:2 | $0.84 \pm 0.03$ | $0.83 \pm 0.01$ | $1.05 \pm 0.05$ | $1.05 \pm 0.07$ |
| C20:3n3 | $0.22 \pm 0.02$ | $0.20 \pm 0.01$ | —— | —— |
| C20:3n6 | $0.17 \pm 0.01$[*] | $0.08 \pm 0.05$ | —— | —— |
| C20:4n6 | $0.10 \pm 0.01$ | $0.15 \pm 0.01$[**] | —— | —— |
| C20:5n3 | $0.64 \pm 0.04$ | $0.84 \pm 0.07$[*] | $1.57 \pm 0.39$ | $2.34 \pm 1.34$ |
| C22:2 | $0.14 \pm 0.03$ | $0.15 \pm 0.00$ | —— | —— |
| C22:6n3 | $0.48 \pm 0.06$ | $0.64 \pm 0.07$[*] | $0.91 \pm 0.25$ | $1.14 \pm 0.23$ |
| **PUFA** | **$26.54 \pm 0.31$** | **$29.10 \pm 0.58$**[**] | **$28.57 \pm 0.54$** | **$30.21 \pm 1.09$** |
| n-3 | $3.17 \pm 0.19$ | $3.90 \pm 0.24$[*] | $4.57 \pm 0.61$ | $5.87 \pm 1.52$ |
| n-6 | $22.40 \pm 0.08$ | $24.22 \pm 0.33$[**] | $22.95 \pm 0.16$ | $23.29 \pm 0.48$ |

**Notes.**

[*]Significant difference ($P < 0.05$).

[**]Extremely significant difference ($P < 0.01$).

SFA, saturated fatty acid; MUFA, monounsaturated fatty acid; PUFA, polyunsaturated fatty acid.

Hepatopancreas is the main organ for lipid storage and lipid processing in crustaceans (*Wang et al., 2008*). A low *HI* was noted after RNAi on *Es-ActRIIB*, and the same result was observed in *E. sinensis* after interference of *Es-MSTN* (*Yue et al., 2020*); a similar phenotype was reported in vertebrates, showing that the inhibition of *ActRIIB* signaling decreases

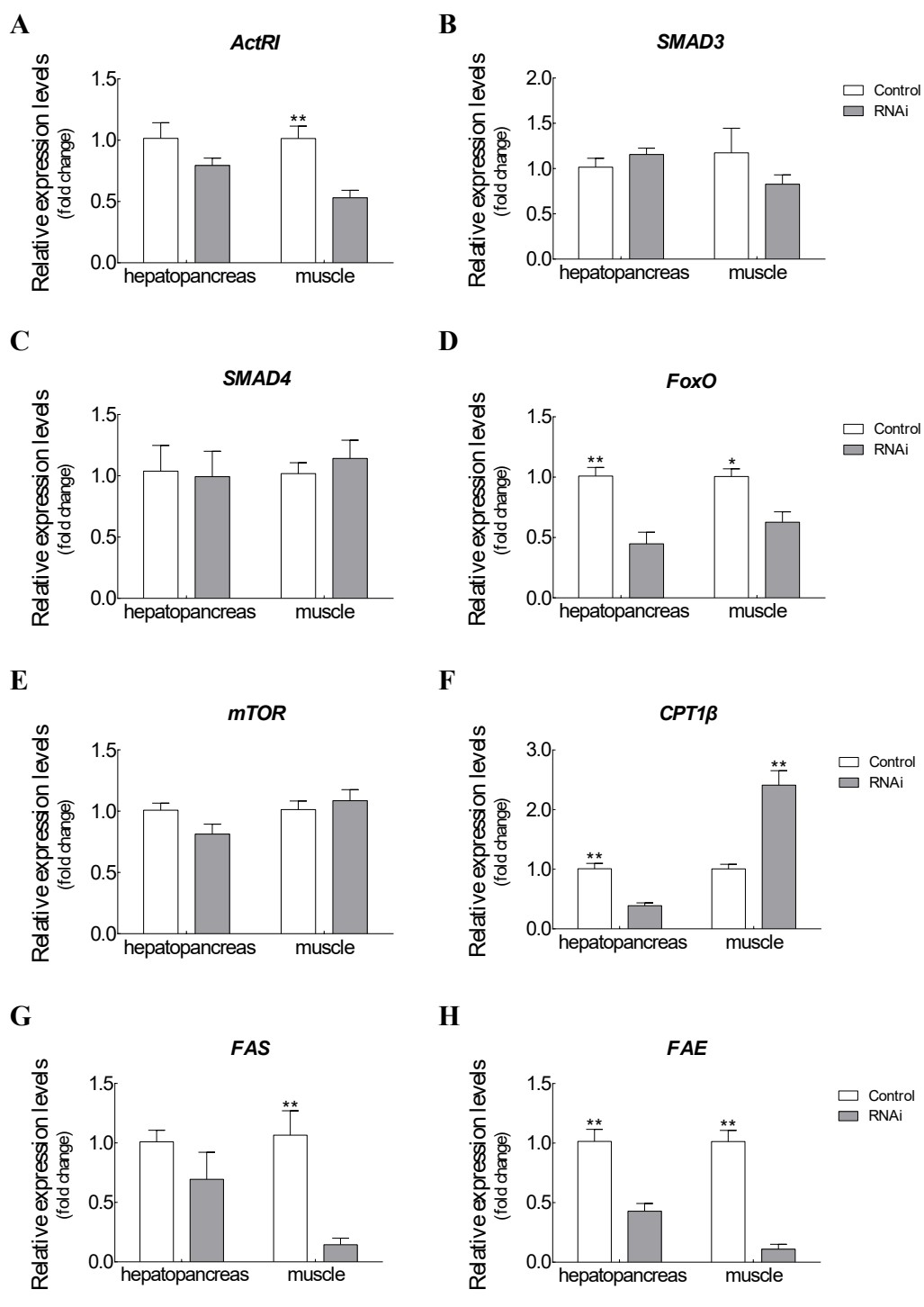

**Figure 5** **Expression changes of target genes after RNAi on *Es-ActRIIB*.** (A) *ActRI*. (B) *S MAD 3*. (C) *SMAD4*. (D) *FoxO*. (E) *mTOR*. (F) *CPT1 β*. (G) *FAS*. (H) *FAE*. "*" indicated significant difference ($P < 0.05$) and "**" indicated extremely significant difference between two groups ($P < 0.01$).

adipogenesis in adipose tissues (*Koncarevic et al., 2012*; *Morrison et al., 2014*). These studies indicated that lipid synthesis could be blocked by inhibiting *ActRIIB* signaling. Accordingly, the fatty acid composition changed significantly in the hepatopancreas, demonstrating that lipid metabolism was affected. Furthermore, *FAE* is responsible for long chain fatty acid elongation (*Igarashi et al., 2019*); the findings showed that low expression of *FAE* in hepatopancreas after *Es-ActRIIB* interference denotes a slowed down lipid synthesis, which was supported by the low *HI*. *CPT1*β is the key enzyme of β-oxidation of fatty acids; the downregulated expression of *CPT1*β in hepatopancreas might imply that energy consumption is reduced in hepatopancreas to support muscle growth, and upregulated expression in muscle indicates that lipolysis is activated and enhanced in muscle to generate additional energy for growth (*Huang et al., 2015*; *Liu et al., 2018*; *Yue et al., 2020*). Moreover, *FAS*, which is responsible for lipid synthesis (*Loftus et al., 2000*), and *FAE* showed downregulated expressions in muscle after interference with dsRNA on *Es-ActRIIB*, indicating that lipid synthesis was inhibited in muscle, and energy was converted to support muscle growth. These results indicate that the balance of lipid metabolism was affected by RNAi of *Es-ActRIIB*.

## CONCLUSIONS

In conclusion, *Es-ActRIIB* is a functionally conservative gene belonging to TGF-β superfamily receptors, and its inhibition could positively regulate muscle growth by affecting protein synthesis and lipid metabolism. This study indicated the negative regulatory function of *ActRIIB* in *E. sinensis*. However, the molecular mechanism of *ActRIIB* signal transduction in *E. sinensis* must be clarified in future studies.

### Funding

This work was supported by Shanghai Agriculture Applied Technology Development Program, China (Grant No. G2017-02-08-00-10-F00076 and No. 2019-3-4), Agriculture Research System of Shanghai, China (Grant No. 202004). The funders had no role in study design, data collection and analysis, decision to publish, or preparation of the manuscript.

### Grant Disclosures

The following grant information was disclosed by the authors:
Shanghai Agriculture Applied Technology Development Program, China: G2017-02-08-00-10-F00076, 2019-3-4.
Agriculture Research System of Shanghai, China: 202004.

### Competing Interests

The authors declare there are no competing interests.

## Author Contributions

- Jingan Wang conceived and designed the experiments, performed the experiments, analyzed the data, prepared figures and/or tables, authored or reviewed drafts of the paper, and approved the final draft.
- Kaijun Zhang, Xin Hou, Wucheng Yue and He Yang performed the experiments, prepared figures and/or tables, and approved the final draft.
- Xiaowen Chen, Jun Wang and Chenghui Wang conceived and designed the experiments, authored or reviewed drafts of the paper, and approved the final draft.

## DNA Deposition

The following information was supplied regarding the deposition of DNA sequences:

The cDNA sequence of activin receptor type IIB is available at GenBank: MN832896 and in Data S1.

## Data Availability

The primers used in this article are available in the Table S1. The GenBank accession numbers of species used in the phylogenetic tree are available in the Table S2. The raw data are available in the Tables S3 to S7. These data were used for statistical analysis of Figs. 3 to 5 and Tables 1 to 3. The cDNA sequence of activin receptor type IIB is available in Data S1 and was analyzed in Figure S1. Some gene sequences used for qRT-PCR are accessible from Data S2.

## Supplemental Information

Supplemental information for this article can be found online at http://dx.doi.org/10.7717/peerj.9673#supplemental-information.

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
