# Peer review of "Molecular characteristic of activin receptor IIB and its functions in growth and nutrient regulation in Eriocheir sinensis"

_PeerJ, doi:10.7717/peerj.9673_

## Round 0.1 · original submission · Major Revisions

Your manuscript need to be deeply revised. Some data have to be included in materials and methods section. For example, weight of specimens or how some samples were taken (more specifications in reviewers 'reports). Besides this, the entire manuscript requires extensive editing for format errors and English grammar.

Reviewer 1 ·

Basic reporting

no comment

Experimental design

no comment

Validity of the findings

no comment

Additional comments

In the manuscript entitled "Molecular characteristic of activin receptor IIB and its functions in growth and nutrient regulation in Eriocheir sinensis", the authors cloned the full-length cDNA of EsActRIIB, and further evaluated the possible role of EsActRIIB in the growth, molting and nutrition regulation with RNAi. Overall, this study is interesting and meaningful. However, there are some drawbacks regarding the materials, results and English writing that seems to be inappropriate to potentially publish the manuscript.

My main concerns are about the introduction, materials & methods, analytical methods:

1. The authors spend a lengthy text describing the role of ActRIIB in vertebrates, but the study on invertebrates is rarely mentioned in the manuscript. The authors indicate that the TGF-β superfamily consists of a large number of structurally and functionally related cytokines subfamilies, however, the authors fail to indicate whether the same family identification relates to the invertebrate ActRIIB receptors. Is the ActRIIB receptor studied in this paper closely related to the members of the mammalian family of receptors?

2. The relationship between the background and study purpose described by the author is very vague, which caused this research seems not to make much sense. Line 116-126: " Unfortunately, in E. sinensis aquaculture, the weight gain rates of crab individuals after molting display significant difference even in the same culture environment, which results in low production of this highly demanded crab…The present research was performed…" this does not seem to be sufficient support for conducting this study. I recommend re-working on the introduction and discussion so that the potential importance of these results might become more obvious.

3. Line 134: "…sampled from four healthy individuals of crab at the four mentioned molting stages", does the authors mean that there is only one individual per molting stages?

4. Fig 4: The authors first examined the distribution of the EsActRIIB mRNA expression levels in 11 tissues (four molting stages), and then analyzed the differences in Mw, Mc and Mp tissues at different molting stages. However, the tissue distribution results show that the EsActRIIB mRNA expression levels in G, I, Tg and Ep are significantly higher. So why didn't the authors choose to analyze these tissues? In addition, I seriously question about the last four results (Fig 4 Hp, Mw, Mc and Mp). Their analysis methods are wrong and unscientific.

5. Table 1: In the control group, the molting interval time of juvenile crabs (4.54 ± 0.85 g) has reached more than 50 days (50.14 ± 7.44). The authors need to explain and clarify what causes the molting cycle to take so long? Are these results scientific? Are these crabs healthy? Are the culture conditions suitable? In addition, the authors need to add the following details in the "Materials Method" section: month, culture conditions, monoculture or mixed culture, etc. These are very important and essential information.

6. In this study, the single molting cycle of juvenile crabs even exceeded 50 days, so I am skeptical of the results and conclusions obtained from this study, as well as their objectivity if the authors cannot explain the reason logically.

Other concerns:

7. The entire manuscript requires extensive editing for format errors and English grammar, and in order for the language to meet with the published standards, it was recommended that some native English speaker should be involved in revising the manuscript. For examples, the abbreviations used for the first time need to coexist with the full name, such as line 42, 53-56. Non-first-time Latin names need to be abbreviated, such as line 59. These are examples taken only from the Abstract. Also, Latin names should be in italics, such as line 411, 435, 439… The authors need to check the entire manuscript.

8. There are no "keywords" found in the manuscript. The authors need to confirm it.

9. The authors describe on line 95 that "blocking ActRIIB signaling reduces fat contents", while " inhibition of ActRIIB signaling enhances obesity and obesity-linked metabolic diseases" on line 99. I wondered about the relationship between ActRIIB signaling and lipid metabolism, and what the authors really wants to express.

10. Line 130: If the authors would like to make a statement about the health of the crabs, they must also indicate how the health of the crabs was assessed.

11. Line 130: The authors need to supplement the crab's weight (mean and standard error) in the manuscript, only "juveniles" is ambiguous and inaccurate.

12. Line 133: Much of the information provided by the authors about sample collection is ambiguous. For examples: Are the "gill" collected by different crabs the same type? As far as I know, the front and back gills seem to play different roles. The same comments can be made for the "walking leg muscle".

13. Line 131-133: The authors divide the molting cycle into premolt (PrM), molt (M), postmolt (PoM) and intermolt (InM) stages, what are the criteria and methods for classification? And (Chen et al., 2017) is not an initial citation, the authors need to verify and provide accurate information.

14. Line 144: What does "he" mean?

15. Was the possibility of genomic DNA contamination taken into account in the total RNA samples?

16. Did the authors sequence the product(s) of their qPCR reactions or products obtained otherwise by using the primers they used for qPCR? What is the length (size) of their qPCR products? The authors detect what they believe to be the EsActRIIB mRNA in all organs/tissues examined but have not determined if the products of their PCR reactions are indeed the EsActRIIB mRNA. The same comments can be made for the reference genes.

17. Line 192: In the "RNA interference conduction" section, why did you choose this injection concentration and volume?

18. Line 223: Is it 48h after the first injection? The authors need to clarify it.

19. Line 313-316: Since the author's data analysis is unscientific (Fig 4), the results obtained are also unreliable. Therefore, this part of the discussion is not credible.

20. Line 344: "…the lipid metabolism…be blocked", does the authors mean that lipid metabolism is not working?

·

Basic reporting

There are small errors (quite a lot) in the writing, some sentences are ambiguous (some of them have been high lightened in the annotated PDF)
Should add some references closely related with this study. Such as:
田志环,焦传珍,成永旭*,吴旭干. 中华绒螯蟹Akt (EsAkt)的cDNA克隆、序列分析及表达特征研究,水产学报,2018,42(4):485-494.
田志环,焦传珍*,成永旭.中华绒螯蟹Smad3 (EsSmad3)的cDNA克隆、序列分析及表达特征研究,中国水产科学,2018,25(2):316-324.
Identification of a transforming growth factor‑β type I receptor transcript in Eriocheir sinensis and its molting‑related expression in muscle tissues, Molecular Biology Reports (2020) 47:77–86.
Zhihuan Tian, Guangchun Lin, Chuanzhen Jiao*.Identification of an S6 kinase transcript in the Chinese mitten crab Eriocheir sinensis, and its molting-related expression in muscle tissues. Fisheries Science, 2019, 85:737–746.

Should add discussion on the nutrient changes in the RNAi groups, especially on the amino acid. what’s the molecular story under this result?

Experimental design

no comment

Validity of the findings

no comment

Additional comments

This study is very interesting. On whether knocking down Myostatin signal in crustaceans will promote or inhibit the animal’s growth, there were conflict results reported before. This study supports the promotion view, it is inspiring on the potential application in Aquaculture.

---

## Round 0.2 · Minor Revisions

Your manuscript still needs to be improved. You should revise the paper in order to avoid ambiguous sentences. Furthermore, the authors should upload the original data.

Reviewer 1 ·

Basic reporting

The revised manuscript looks very messy, which makes it difficult for me to find and
review the author's changes in the text. The author needs to highlight the revised
content in yellow and indicate the line number, rather than the current form.

Experimental design

See comments for the author

Validity of the findings

See comments for the author

Additional comments

Although the authors have responded to the reviewer's comments, there are still some
comments that need to be corrected before publication:

Line 183: "...we collected the relative tissues of four crabs..." This sentence is still
ambiguous.
Regarding previous question "The authors first examined the distribution of the
EsActRIIB mRNA expression levels in 11 tissues (four molting stages), and then
analyzed the differences in Mw, Mc and Mp tissues at different molting stages.
However, the tissue distribution results show that the EsActRIIB mRNA expression
levels in G, I, Tg and Ep are significantly higher. So why didn't the authors choose to
analyze these tissues?", the author's explanation is far-fetched. According to the
author's response, I would like to know what was the purpose of the author's initial
examination of the EsActRIIB mRNA expression levels in 11 tissues?
In addition, the analysis method seems to be incorrect and unscientific despite the
revision made by the authors (Fig 3 Hp, Mw, Mc and Mp), and it is suggested that the
authors discuss the analysis method carefully. If possible, the authors should upload
the original data.

Annotated reviews are not available for download in order to protect the identity of reviewers who chose to remain anonymous.

---

## Round 0.3 · Minor Revisions

Your paper has been improved. However, there is one important think that has to be modifed. It is needed to normalize the relative gene expression (e.g. according to Vandesompele, et al. 2002) of real time quantitative RT-PCR.

Reviewer 1 ·

Basic reporting

non

Experimental design

non

Validity of the findings

non

Additional comments

As I had mentioned before, the method the authors used to normalize the real-time quantitative RT-PCR data is a single internal control gene method (known from the reference the authors cited). However, the real-time quantitative RT-PCR data should be accurately normalized by geometric averaging of multiple internal control genes due to the multiple internal control genes (β-Actin, S27, and UBE) were used in the present manuscript. The authors should revise the calculating method and calculate the relative gene expression by Vandesompele, et al. (2002).

In addition, the authors described in the text that the relative expression level of target genes were estimated by the 2-ΔΔ△△Ct method according to the Livak & Schmittgen (2001). However, when I reviewed the original data, I found that the authors did not use this method, therefore, the authors need to check and correction.


(Vandesompele J, De Preter K, Pattyn F, et al. Accurate normalization of real-time quantitative RT-PCR data by geometric averaging of multiple internal control genes[J]. Genome Biology, 2002, 3(7): 1-12.)

---

## Round 0.4 · accepted · Accept

Thank you for improving your paper as indicated. I am pleased to inform you that now your paper has been accepted for publication in PeerJ.
Thank you for sending your work to this journal.